# A Review of the Environmental Benefits of Using Wood Waste and Magnesium Oxychloride Cement as a Composite Building Material

**DOI:** 10.3390/ma16051944

**Published:** 2023-02-27

**Authors:** Dorin Maier

**Affiliations:** Faculty of Civil Engineering, Technical University of Cluj-Napoca, 400114 Cluj-Napoca, Romania; dorin.maier@ccm.utcluj.ro

**Keywords:** wood waste, magnesium oxychloride cement, sustainability, waste utilization, environmental benefits

## Abstract

There is an increasing awareness of the negative environmental impact produced by human activity worldwide. The scope of this paper is to analyze the possibilities of the further use of wood waste as a composite building material with magnesium oxychloride cement (MOC), and to identify the environmental benefits offered by this solution. The environmental impact of improper wood waste disposal affects both aquatic and terrestrial ecosystems. Moreover, burning wood waste releases greenhouse gases into the atmosphere, causing various health problems. The interest in studying the possibilities of reusing wood waste increased significantly in recent years. The focus of the researcher shifts from considering wood waste as a burning fuel to generate heat or energy, to considering it as a component of new building materials. Combining MOC cement with wood opens the possibility of creating new composite building materials that can incorporate the environmental benefits offered by the two materials.

## 1. Introduction

Societies around the world are in continuous development and according to late estimates, 60% of the world’s population will reside in urban areas by 2030. Even though they only cover 3% of the planet’s surface, cities consume 60–80% of the world’s energy and produce 70% of their carbon emissions [1]. The 2030 Agenda for Sustainable Development [2] not only promotes all efforts to make cities and industries more sustainable, and also promotes the implementation of consumption patterns and goods that can advance economic success and good health while preserving the environment [3]. The cement and concrete industry are at the core of efforts to enhance the sustainability and resilience of the urban environment by promoting population expansion through green buildings, since it forms the foundation of current and future cities [4].

Many businesses, including the construction industry, are developing policies to lessen their environmental impact as a result of rising environmental consciousness worldwide [5]. Governments must also cut back on waste and emissions. For this reason, the construction industry is making great efforts to limit the use of energy and natural resources in buildings, and material producers are launching sustainable initiatives to accomplish these goals. In this regard, the development and production of more environmentally friendly and sustainable building materials is a contemporary trend in the construction industry, which aims to safeguard and reduce the effects on the environmental [6].

In recent years, the relevance of renewable building materials has significantly expanded as a result of the increased awareness of the environmental problems caused by the construction sector. One of these is wood, and there is an increasing amount of interest in learning about and creating innovative ways to employ it in the construction process [7].

Throughout history, wood has been used by humans to meet a variety of demands, including providing building materials for shelters, the tools for agricultural or hunting activities, and fire to cook or heat the living space. Up until the second half of the nineteenth century, wood was the primary material utilized for both construction and energy production. In terms of its use in architectural construction and design, wood is one of the more traditional materials, yet it is by no means an outdated one. Steel and concrete lack certain qualities that wood possesses, and new wood construction techniques are always being offered to the market [8]. Logging, sawmilling, making plywood, fiberboard, and chipboard from wood, making furniture, paper, and matches, and producing diverse wood products such as tool handles, sporting goods, weaving equipment, and wooden toys are all examples of wood-based industries [9].

Wood is a material with great mechanical and thermal properties that are naturally occurring, renewable, and biodegradable. Wood materials typically have a substantially lower environmental impact during the production and disposal phases than comparable products made from inorganic or fossil source materials [10]. In addition, wood does not compete with food-like resources having an agricultural origin [11]. As a result, since the start of the twenty-first century, there has been a growth in the consumption of wood for new applications in addition to its conventional usage (energy production, building materials, chemicals, etc.) [12]. A study shows that by 2030, wood production could be insufficient to meet the demand in Europe [10]. Along with an increase in wood use, end-of-life wood-based products are producing more wood waste. Thus, recycling this substantial deposit could serve as a cheap, plentiful supply of raw materials to produce new products.

It is not feasible to completely replace concrete buildings with wood structures because each building material has benefits and drawbacks. Combining the environmental benefits of wood with the benefits of concrete to produce a more sustainable building material is a better strategy. This can be done by reducing the amount of cement in the concrete mix and replacing it with crushed wood pieces. This solution implies the use of small pieces of wood, so in this mix, wood waste can also be used. Since it is well known that waste management is a significant problem in the construction industry, developing a solution to further use a material considered as waste fits perfectly in the circular economy principles.

Large amounts of waste are produced by the wood industry, which must be used, traded, or properly disposed of. The wood business encounters wood leftovers frequently all year long, but even if they need to meet the demands of bioresources and energy, these resources are not fully utilized; thus, the management of wood waste is poor. The need for wood waste for energy production does, however, fluctuate according to the summer and winter peaks [13]. The incinerator plant’s processing capacity decreases as a result of the summertime reduction in energy production. As a result, there would be more wood waste accessible for use as a raw material by the wood processing sector [14].

The incompatibility of wood and cement is the main obstacle to the development of wood-cement building materials [15]. This results from some of the soluble compounds in wood preventing or slowing the cement’s hydration process [16,17]. As a result, compared to products made of ordinary cement, wood-cement composites have lower mechanical strengths. Utilizing different substitutes, such as cement based on magnesia, in wood-cement composites are one way to overcome this incompatibility [18]. MOC cement may be the ideal choice for wood-cement products due to its low alkalinity and quick setting time. The MOC cement’s natural hue is yellowish and closely resembles the color of several types of natural wood [19].

The need for a greater understanding of the use of wood waste and MOC composite as a building material is highlighted by the overly general backdrop that was briefly given. When discussing this subject, many questions can be asked, such as: Can this material replace concrete or steel? Why is it not widely used? Does it resist in time? The list can go on. Most of the questions have as the main base the environmental benefits that can be brought about by the use of the new composite material compared with the traditional and more energy-demanding building materials. In this context, the main question addressed in this study can be formulated as: “*What are the environmental benefits of using wood waste and MOC composites as building materials*?”

The review procedure includes a thorough analysis of the literature using scientometric analysis, in-depth analysis, as well as input from a wood expert. The Clarivate Web of Science database provided the bulk of the data for this study, which were then extracted using the PRISMA (Preferred Reporting Items for Systematic Reviews and Meta-Analyses) technique.

The paper is formatted according to the purpose of this study. The main scope is to offer an overview of the main environmental benefits generated from the possibility of mixing wood waste and MOC cement to develop new composite building materials. In this sense, after the introduction section, the following sections of the paper provide a short presentation of the two materials considered in this study—wood waste and magnesium oxychloride cement (MOC). The materials and methods are presented next, followed by a section containing the main scientometric analysis results. The in-depth analysis of the literature is presented in two parts. One part is reserved for the overview of the main knowledge regarding wood waste—MOC composites as building materials—and the other part is dedicated to the identification of the main environmental benefits of the composites. The paper ends with the presentation of the main conclusions of this review paper and the main references used.

## 2. A Short Background of Wood Waste

The term “wood waste” refers to any unwanted or discarded materials that are produced during the processing, manufacturing, or use of wood products [20]. Wood waste may also refer to wood products that have reached the end of their useful life, such as wooden pallets or shipping crates, and are no longer needed or suitable for their original purpose [21].

From a terminology point of view, a certain difference in terminology needs to be pointed out. Throughout the literature, there is a change in the order of the words “wood” and “waste” to form the term “waste wood”, which can be found in [22] or [23,24]. Although the terms “waste wood” and “wood waste” are often used interchangeably, there is a subtle difference in their meaning depending on the context in which they are used. “Waste wood” generally refers to wood that is generated as a byproduct of a manufacturing or processing process and is not suitable for reuse or recycling, while “wood waste” is a more general term that encompasses any unwanted or discarded materials that are produced during the processing, manufacturing, or use of wood products. So, when talking about a broader range of materials that may still have some potential value or use, the term “wood waste” must be used. This is also the case of this paper.

Wood waste can be a valuable source of material for energy generation [25] as well as producing wood-based products such as particleboard, pulp and paper, and composite materials [3]. However, wood waste can also pose environmental and health hazards if it is not properly managed and disposed of. To have an image about the size of the phenomenon in [26], it is stated that the quantity of wood waste, at the European level, is around 33.20 million tons. The same source presented that the geographic distribution of the quantities varies between countries and regions. The largest amount of wood waste, estimated to be 110 kg/inhabitant/year, is estimated to be in the Nordic countries, which can be explained by these countries’ well-known wood tradition. The smallest amount of wood waste, which is half of that produced by the Nordic countries, can be found in the southern and eastern countries, with an estimation of 55–60 kg/inhabitant/year. The western countries report an amount of wood waste up to 75 kg/inhabitant/year.

There are three main treatment methods used to dispose the amount of wood waste: the most common method is disposal, such as landfill and incineration, which accounts for around 37% of the total waste; material recovery and transforming the waste, often into particleboards, which accounts for around 33%; the least common method is energy recovery, such as generating heat or cogeneration, which is used for 30% of the total wood waste [26]. The difference in approaching the disposal of wood waste can be observed in different regions of Europe. While a great deal of wood waste is buried in most of Europe, in northern and western countries, the recovery process is given more importance.

An increasing tendency can be observed in finding possibilities of further using the wood waste and developing new products. To better understand how wood waste may look like, Table 1 presents a classification of wood waste according to its origin, as found in [10]. The researchers presented the origin, type, and class of wood waste. The classification of wood waste in one of the fourth classes is done according to its capacity of being reused, mainly from the structural strength of the material.

A wood waste characterization analysis was carried out in order to better comprehend the wood waste and how it can be preserved in the next section. Waste characterization is a crucial component of any waste management and entails examining the various compositions of the waste stream. To fully treat trash, creators of innovative waste technologies must take into account the fundamentals of waste streams [27].

According to the study of [9] the composition of wood waste usually consists of the parts presented in Figure 1:-Bark: the outermost layer of woody plants, which overlaps the wood (Figure 1a);-Sawdust: produced during wood processing, wood scraps (Figure 1b);-Chips: wood waste that results from smoothing or planing wood Figure 1c,d);-Wood cutouts: medium-sized solid materials made by sawing or sawing large pieces of wood (Figure 1e);-Wood rejects: wood/timber that has been rejected either due to decay or pest infestation (Figure 1f).

Wood waste can be classified according to condition and category based on its quality:-Clean wood waste includes leftover pieces of concrete that are less than 2 inches in diameter, such as scrap lumber, sawdust, plywood, and wood used in concrete. There may be some respectable quantity of bolts, screws, or nails in clean wood. Telephone poles, treated wood, tar wood, and other materials are not considered clean wood trash because they include paint, oil, or Styrofoam. Note: construction and demolition trash that contains paint or other pollutants can be recycled.-Contaminated wood waste is treated wood in general, including painted or coated wood, building wood, furniture, and roof boards.

The classification of wood waste can also be made by considering the level of wood contamination. As shown in [28], this classification implies the use of the following grades:-Grade A: in this category, pieces of wood have no or very few contaminants, such as paints or other layers of protection liquids.-Mixed grade: in this category, the level of contamination is increasing, and small amounts of concrete or nails are present on the surface of wood waste.-Low level: this is the category where pieces of wood that are very contaminated are included.

The possibility of using wood waste is done according to each grade class. For the materials, recycling wood waste from grade A is more suitable, as it can be easily cleaned and reused either as building material for temporary structures or in the process of panel production. The wood waste from the mixed grade can also be cleaned and reused, mostly by grinding and using it as raw material for panels, whereas the wood waste from the low grade, which cannot be cleaned in most cases, will be disposed in landfills [28].

## 3. A Short Background of Magnesium Oxychloride Cement (MOC)

More than 150 years ago in 1867, a French civil engineer, inventor, and chemist by the name of Stanislas Sorel discovered Magnesium Oxychloride Cement (MOC) [19]. MOC cement is also known as Sorel cement because of the name of the inventor [29]. MOC is a unique type of cement that sets quickly, binds strongly, and has high strength in normal air conditions [30].

MOC cement has received much interest because, among the low-carbon cement materials, it has significant potential to replace Portland cement and, in many instances, demonstrates qualities superior to those of ordinary Portland cement [31]. The key benefits that distinguish MOC cement from regular cement include reduced CO_2_ emissions, rapid setting, absence of humid curing, low alkalinity, and great mechanical strength [32].

MOC has been successfully used as environmentally friendly fireproof thermal insulation products, urban refuse/cement compound floor tiles, and light-weight boards [33] mainly due to its superior properties, such as high strength at an early stage, outstanding thermal insulation [34], fire resistance [35], and distinguished resistance to abrasion [36,37].

MOC cement offers several attractive cementitious substrates since the phase composition could be controlled by modifying the precursor’s molar ratio [38,39]. Additionally, it has rheological characteristics that allow the substance to flow into crooked cavities [40]. Additionally, it demonstrates relatively high elastic modulus (70–85 GPa), compressive strength (69 MPa), and flexural strength (4 MPa) [41].

Despite these beneficial qualities, MOC is not widely used since it rapidly loses strength when exposed to water over an extended period of time [42]. Several techniques have been used to improve the water resistance in order to increase the application of MOC, but the most successful one is the addition of additives and mineral admixtures [43].

Although additives are successful in improving the qualities of MOC, all attempts to address the drawback of low water resistance entail the addition of various additives in the mixture [44]. However, this significantly increases the cost of the end materials [45]. Therefore, MOC cement must be combined with other affordable materials, as well as general trash from other sources, in order to create a more sustainable and cost-effective building material.

Industrial flooring, fire protection, grinding wheels, and wall panels are the main commercial applications for MOC cement [46]. It is appropriate for usage in construction projects that require limited time for execution, such as restoring infrastructure such as a highway or an airport runway, due to its rapid setting and hardening [17]. The MOC cement has the benefit of decreased alkalinity, which makes it an excellent choice for usage with glass fibers without the aging issue that is frequently present when glass fibers [47] are combined with regular cement. To create composites and building materials that resemble wood, MOC cement works well when combined with sawdust and wood shavings [17].

In comparison to gypsum or fiber-based boards, MOC boards have several superior qualities, such as improved resistance to abrasion, lower thermal conductivity, higher strength, and stronger fire resistance. Numerous studies have been conducted on product characterization and development, including to increase MOC water resistance, as a result of the adaptability of magnesium-based cement building materials [48].

## 4. Materials and Methods

According to the purpose of the paper, the needed data were obtained by interrogating one of the most important scientific databases using “wood waste” as a search keyword, before processing the data using specific software. The working plan of these methods includes the selection of articles in a sample database followed by filtering and refining the bibliographic data.

### 4.1. Data Collection

The database used to obtain the data for this study was Calarivate Web of Science Core Collection (WoS). The process of interrogating the database is illustrated in Figure 2. The keyword used for this study was “wood waste” and the main filters to limit the results were related to the selection of only article or review paper types to be included in the Civil Engineering field of research. The interrogation of the WoS database by using “wood waste and MOC cement” as keywords, and maintaining the same filters, resulted in few papers that were already included in the “wood waste” search. In this sense, the data collection section and the entire study, even if it focused on papers having “wood waste” as a topic of research, also includes the main study regarding the use of MOC cement with wood.

As can be observed, the initial results indicate almost 13,000 papers related to the topic of “wood waste”. After the application of the first filter, only article and review paper types, the number decreased to almost 2000 papers. The last filter was the selection of papers grouped in the civil engineering field of research and the number of papers decreased to a more manageable number—328.

In the process of selecting the suitable papers for this review, the guidelines of PRISMA method was used. PRISMA (Preferred Reporting Items for Systematic Reviews and Meta-Analyses) is a data identifying and extracting method proposed by Moher et al. [49] that implies the use of four steps: identification, screening, eligibility, and inclusion.

### 4.2. Data Extraction and Analysis

The process of scientometric analysis implies the extraction of raw data from the scientific database in a format suitable for the main software used. In this case, the software used for this review was Bibliometrix (version 3.1), developed by Massimo Aria and Corrado Cuccurullo, Department of Economics and Statistics, University of Naples Federico II, Italy [50].

The input data necessary for the software were exported in a plain format file and contained the full range of resources available in the WoS database. Besides the title, authors, and abstracts of the papers, the plain format file contains a large range of data such as the author keywords, keywords plus, the authors’ affiliation, the journal names, and the reference lists. The limitation of using bibliometric software is that there is a need for a certain standardization of data, so before any results were generated, a significant portion of time was dedicated to small corrections needed in the input file.

## 5. Results and Interpretation

### 5.1. The Evolution of the Annual Number of Published Articles

The interest in a certain subject can be seen in the number of papers published annually having that subject as the topic. To illustrate the evolution of the annually published number of papers having “wood waste” as a topic, a graph was generated (Figure 3) based on the data extracted from the WoS database.

The grouping of the articles according to their publication years was taken directly from the WoS database, without a detailed verification of the publication year. The graph indicates on the x-axis the years of publication and on the y-axis the number of articles published each year.

The graph from Figure 3 indicates an important increase in the interest in the use of wood waste in the civil engineering field in recent years. The first year with a paper indexed in the WoS database is 1991, and since then, up until the year 2013, the number of papers published annually was very little, with a peak of 10 papers published in 2007. In 2013, 14 scientific works were published, and since then the annual number of published articles has varied, maintaining over 10 papers per year, reaching 18 papers in 2016, and then decreasing to 14 papers in 2017. A significant increase can be observed starting with the year 2018 when 36 works were published, and then even if only 29 papers were published in 2019, the interest in this subject increased substantially, reaching 45 papers in 2020 and 70 papers in 2021.

### 5.2. The Trend Topic in Wood Waste Research

The evolution of the annual number of published articles dealing with wood waste as a subject reveals an increased interest in recent years. For a better understanding of the main subject approached by the researchers, a trend topic graph is presented in Figure 4. The graph was generated with the help of Bibliometrix software based on data extracted from the WoS database. The trend topic was constructed using the keywords used by the authors with a minimum word frequency of five, and the number of words considered each year was set to three. The graphic representation of the analysis performed by the software consists of a series of lines and bubbles. The lines represent the period of using that term and the bubbles represent the term frequency. The size of the bubble indicates a certain level of frequency for each term; the bigger the bubble, the more frequently used that term is.

The data presented in Figure 4 reveal an evolution of the main keywords used by the researchers in their studies. In the early studies, the most used terms were wood, sand concrete, or lightweight concrete, and this can indicate that wood waste was considered as a solution to decrease the weight of the concrete by adding grinding wood to the concrete mixture. Another direction of research focused on using wood waste in the composition of particleboards, as filler material or even as a thermal insulation material. In recent years, the most frequently used terms, where the bubbles are bigger, refer to words like mechanical properties, durability, strength, and compressive strength, which indicates that a big part of the research is focused on the mechanical properties of the wood waste. In this way, new building materials can be made using wood waste. The last approach is presented in the context of increasing the importance of environmental issues; subjects such as recycling or circular economy are becoming more present in construction research topics.

As observed in the trend topic graph for recent years, there has been a growing interest in the research of wood waste and its potential as a valuable resource. This interest is driven by a variety of factors, including environmental concerns, the need for sustainable and renewable energy sources, and the desire to reduce waste and increase resource efficiency. There is a clear direction in the development of new materials made from wood waste, such as composite materials and bio-based plastics. These materials have the potential to be both more sustainable and more cost-effective than traditional materials, while also providing new opportunities for waste reduction.

Overall, the increasing interest in wood waste research reflects a broader shift toward a more sustainable and circular economy where waste is minimized and resources are used more efficiently. As such, it is likely that this research will continue to be an important area of focus in the years to come.

## 6. Composite Building Materials Made of Wood Waste and MOC

The wood-cement composites are relatively new building materials and there are many unknowns around them. Since the results obtained so far indicate superior external qualities, wood-cement composites are now being studied and produced industrially in several parts of the world, mostly in the form of panels [51]. The chemical incompatibility between wood and cement, which is typically just plain Portland cement and prevents cement from setting and hardening, is a fundamental challenge for manufacturing wood-cement composites [16]. Several sugars, a portion of hemicelluloses, and their breakdown products are the primary inhibitory chemicals. The wood species, location, the section of the tree, time of year when the wood is cut, wood/cement ratio, type of cement, storage conditions, and other factors all have an impact on the inhibitory degree [52]. Compared to regular Portland cement, the wood had a lessening influence on the hydration of magnesium oxychloride cement; thus, the use of MOC cement in the composition of wood cement composite material can be a solution [28].

The researchers focus their attention on solutions of combining wood with concrete, especially the further use of wood waste. A direction was the use of composite wood waste and concrete building materials to produce wall panels and hollow blocks mainly as a thermal insulating solution [53]. The recycling possibility of wood waste is still in the early stage of research, but promising results have been obtained by researchers [54] by using it in a wood cement composite for wool cement board productions. According to studies, wood-MOC composites made with a higher wood fiber content showed lower thermal conductivity, higher bending resistance, higher residual bending resistance after being exposed to high temperatures and water immersion, and a better effect on noise reduction [55].

The feasibility of the wood-MOC composite building materials is given by the properties of MOC cement to act as a binder for the wood fiber. The researchers in [51] proposed to use it as a reliable and environmentally friendly binder for plywood applications.

In the study [33], MOC cement deposited on the surface of the fibers was connected by the hydrogen bonds in wood, improving the mechanical properties of wood-based composites. The study focused on the preparation of an eco-friendly and high-performance MOC-based formaldehyde-free adhesive based on an organic-inorganic hybrid structure. Additionally, the plywood’s flame resistance was enhanced by the application of MOC. The good properties of wood-MOC composites were highlighted also in [17], where the authors identified that the specific dry density of the composite material is near 1.0, and behaves like hard natural wood when nails are driven into it.

Wood can be used in the mix in the form of ash; in [56] the wood chips ash “greatly improved the mechanical strength of the lightened composites and enabled the development of construction materials with lower environmental damage”. The beneficial effect of the fly ash in the MOC mix are presented in [57] where the research results “indicate that fly ash promotes the durability of MOC-solidified sludge, and the fly ash-MOC blend is proved a profitable and sustainable material for sludge solidification”.

The main disadvantage of wood-MOC composite material is the same as the main disadvantage of MOC cement—poor water resistance. In the test performed in [58], the building components were submerged in water, and it was discovered that two months later, the magnesium oxychloride cement specimen had suffered substantial quality loss, with a more than 80% reduction in compressive strength. Other drawbacks such as moisture absorption, efflorescence, or corrosion of the reinforcement in the reinforced concrete building components result from this poor water resistance. The poor water resistance of MOC cement has prevented the further commercialization of this product [33].

### 6.1. Possible Solutions to Overcome the Limitations of Wood-MOC Cement Composites

Given the multiple benefits of MOC cement, several researchers focused their attention on finding solutions to improve the poor water resistance of the cement, obtaining encouraging results. For instance, in [59] the authors shows that a tiny amount of phosphoric acid or soluble phosphates can significantly increase MOC’s water resistance. In other studies, the use of additives proves to be a good solution to overcome the water solubility issue of MOC cement [60]. Another category of solutions is related to the curing process of the cement. The inclusion of high-temperature curing at 75 °C can significantly increase the compressive strengths of these MOC cement-based composites that have undergone ambient curing at an early age, as demonstrated in [61]. The H_2_O/MgCl_2_ mole ratio in the MgO-MgCl2-H_2_O ternary system was closely related to the curing temperature on the phase structure and mechanical performance of the MOC cement, according to an analysis of the curing process and its effects [62]. In the case of wood-MOC paste, by adding supplementary cementitious materials, higher strength retention compared to the control samples was obtained [63].

In [63], researchers showed that adding supplementary cementitious materials, especially incineration sewage sludge ash (ISSA), could improve the water resistance and volume stability of the wood-cement paste. The water resistance of the MOC composites is also improved by using biochar, as shown in [64]; the researcher demonstrated that “the naturally porous structure of biochar can form a protective core-shell structure with hydration products, which not only contribute to improving the stability of phase 5 and facilitating stress dissipation during mechanical loading, but also alleviate the filtering effect of wood”.

Encouraging results were obtained in [65] where, by adding an organic and inorganic hybrid of tannic acid in the mix of MOC, there was an improvement in the water resistance and compressive strength. This demonstrated better results in both cases compared to other solutions where in order to improve the water resistance, the compressive strength was usually sacrificed.

The research results presented in [66] highlight the beneficial effect of using chemical additives, such as “phosphoric acid (3%), citric acid (4%) and urea–formaldehyde resin (12%)”, to the compressive strength and water resistance of MOC solidified sludge. The efficient use of ferrous sulfate heptahydrate, phosphoric acid, and polypropylene fiber for the improvement of MOC water resistance and compressive strength is shown in [67].

The additives are efficient at improving the qualities of MOC in an effort to mitigate the drawback of low water resistance, but the cost of the finished product is significantly raised [44]; thus, the use of waste in the composition of the composite material is a good solution to decrease the costs. The researchers proposed the solution of using low-cost FeSO_4_ as good water resistance in [45]. Their results indicate that “the addition of 2% FeSO_4_ made the softening factor of the F-BS/MOC reach 0.82 after 28 days, and the water absorption rate was reduced to 9.6%”.

Although many benefits and good properties appear by using the wood-MOC composite as building material there are a lot of parts of it that are still unknown, as shown in [68], as even if excellent mechanical, chemical, and biological qualities are displayed by the material, there is not much knowledge about its mechanical behavior, yet. Finding the right balance in the wood-and-cement combination is important, but it can be challenging so this subject must be further studied and more solutions should be developed.

### 6.2. The Mixing Proportions of Wood Waste and MOC Cement

When talking about mixing proportions between wood waste and MOC cement there are a lot of elements that need to be considered so the mixture can vary depending on the specific application and desired properties of the composite material. Being a composite material still in development, there is not yet a final formula for the mixture proportions. From the wood waste point of view, it is important to know that different types of wood waste, such as sawdust or shavings, have different properties that can affect the final composite material. The most important ones are the *particle size* and the *desired properties*, such as strength, durability, or insulation.

In the study of [69], the physico-mechanical properties of magnesium oxychloride cement board were investigated, and their results show that incorporating wood fibers in the mix led to the maximum increase of the toughening effect and the flexural strength if the content of wood fibers is 75% (by weight of MgO), but if this percent is exceeded, the strength decreases substantially. An increase in the flexural strength of wood-cement boards with the increase of the wood fibers was also observed by researchers in [16]. In their study, the boards were prepared with 15%, 20%, and 25% (% by mass) wood fibers, obtaining an increase in the flexural strength from 11.4MPa, the 15% boards, to 18.9 MPa, the 25% boards.

As a general impression, a typical mix might consist of 50–70% wood waste and 30–50% magnesium oxychloride cement. However, in future research, laboratory tests will be performed so more exact proportions that yield the best results for a specific application can be obtained. Additionally, a thorough understanding of the properties of both the wood waste and the magnesium oxychloride cement is necessary to ensure that the final product meets the desired specifications.

## 7. The Environmental Benefits of Using Wood-MOC Composite as a Building Material

By developing a composite material, the properties of the component materials are combined so in order to analyze the environmental benefits of wood-MOC material, the properties of wood wastes and MOC cement must be analyzed. As was shown in the sections above, the solutions to counteract the disadvantages of MOC cement imply a cost increase for the final product so the use of waste in the mixture will lead to a balance in the production costs. In the same way, the environmental benefits of MOC cement are enhanced by those of wood waste.

From the MOC cement point of view, the main environmental benefits, compared to the ordinary Portland cement can be summarized as: *low CO_2_ emissions*—magnesium cements have a lower carbon footprint compared to traditional Portland cement, as they require less energy to produce and generate fewer greenhouse gas emissions [70]; *recyclability*—magnesium cements can be easily recycled, as they are made from natural and readily available raw materials, and the manufacturing process produces minimal waste [71]; *sustainable raw materials*—magnesium cements are made from abundant and sustainable raw materials, including magnesium oxide and magnesium hydroxide, which can be derived from a variety of sources, including sea water, mineral deposits, and industrial waste [72]; *durability*—magnesium cements have a long service life and are resistant to environmental degradation, reducing the need for maintenance and replacement over time; *improved indoor air quality*—magnesium cements have been shown to improve indoor air quality by reducing the emission of volatile organic compounds (VOCs), which can be harmful to human health [73]; *lower water demand*—magnesium cements have a lower water demand compared to traditional Portland cement, reducing the amount of water required for construction and minimizing the environmental impact of water scarcity [74,75].

The benefit of reducing the greenhouse gas emissions in wood panel productions by using MOC cement is highlighted in [16] where the researchers identified that the manufacturing of wood-MOC board using cremated sewage sludge ash resulted in greenhouse gas emissions that were 71% lower than those of plywood and comparable to resin-based particleboard. Additionally, the production of conventional resin-based particleboard used a lot of very toxic organic resin, but the creation of wood-MOC board used 58% less.

The environmental benefits of further use of wood waste are given by the fact that wood may be recycled and reused to conserve natural resources. The reuse, recycling, and disposal of wood waste allows forests to recover and allows for the replacement of harvested trees, which relieves strain on them. Forests are overexploited and occasionally destroyed as a result of this practice because of the great demand for wood and the products derived from it.

The most frequent disposal method for wood waste involves incinerating it, and because this is done outdoors, dangerous chemicals like carbon monoxide, sulfur dioxide, nitrogen oxides, and ash are released into the atmosphere. The smoke from the careless burning of wood debris contains tiny particles that might damage the lungs. Dioxin and polycyclic aromatic hydrocarbons (PAHs), two compounds found in wood smoke, are thought to be carcinogenic. Wood smoke hinders children’s proper lung development and raises the risk of lower respiratory tract infections. Additionally, it might weaken the immune system and make breathing challenging.

Dumping wood trash in neighboring open spaces is another typical method of disposal. Wood trash starts to disintegrate when left unattended over time and emits methane gas, a dangerous greenhouse gas. Waste production is a necessity of existence; it cannot be prevented, but it must be managed. Waste causes a variety of issues, including air pollution, environmental damage, loss of aesthetic value, the production of offensive odors, and loss of aesthetic value due to inappropriate combustion of waste. Waste can be harmful to your health if not disposed of properly; poorly discarded waste serves as a breeding ground for pests and disease carriers. In addition, waste (such as sawdust) dumped into water bodies can obstruct drainage and result in flooding during the rainy season, causing damage to property and loss of life.

The benefits given by the use of wood waste can be reflected in the management of construction and demolition waste. In the case of construction waste, since the main impurities that can carry this type of wood are nails and other metal components that are removed by magnets, wood waste corresponds to untreated or simply physically treated wood, so no specialist sorting equipment is needed. Additionally, a manual inspection sorting process is used to separate any other materials that might have been mixed in with the wood debris, such as fragments of plastic or paper. This garbage is then put through a particle size reduction process according to customer requirements [76]. However, larger sorting percentages (up to 100%) are preferred because the inclusion of other types of materials alters the chemical characteristics of recycled wood waste (e.g., metals and plastics). Furthermore, tainted wood debris might harm machinery. This sparked research into more specialized tools for the identification and grading of heavy metals in wood waste up to this point [77].

The use of wood-MOC cement composite material can also help the environmental problems in the wood industry, especially related to the processing of wood. Here, waste is mainly generated by the processes of transforming the wood into derived goods to meet customer use expectations, like furniture, plywood, or sawmills. Bark, debarking cores, veneer waste, and panel trim are the main sources of wood waste in the plywood sector, whereas chipboard waste is extremely minimal but comes from chip waste from panel cuts and dust from grinding machines [78].

## 8. Conclusions

The entire society is becoming more aware of the environmental impact caused by the complexity of human activities. Waste management is one of the research directions that received great attention from the scientific community due to its immense potential to benefit the environment. From the construction industry’s point of view, the largest potential for reusing and prolonging the life of materials is wood waste.

The interest in finding possibilities of reusing wood waste has increased substantially in recent years. If wood waste was reused just as fuel to generate heat or energy in the past, lately the focus has been on understanding and testing its properties and using it in the composition of new building materials. The possibility of using wood waste in the concrete mixture was considered for a long time but the incompatibility between ordinary Portland cement and wood led to limitations in research. Most recently, the mixture of MOC cement with wood provided encouraging results, thus opening the possibility of obtaining new composite products that can lead to a decrease in the environmental impact of the construction industry.

The main environmental benefits of using wood waste and magnesium oxychloride cement composites as building materials are: *waste reduction*—the use of wood waste as a component in building materials reduces the amount of waste that would otherwise go to landfills; *renewable resources*—wood waste is a renewable resource; *energy efficiency*—wood waste composites can provide insulation and help to reduce heating and cooling costs, making buildings more energy-efficient; *carbon sequestration*—wood waste is a carbon sink, meaning that it can store carbon dioxide, which helps to mitigate the impact of greenhouse gas emissions; *low-emission manufacturing*—the production of magnesium cement typically generates lower levels of greenhouse gas emissions compared to traditional cement production.

Being a relatively new material, there are some limitations despite its multiple environmental benefits, but these may be overcome if more studies and research are conducted. One of the main limitations of the wood waste—MOC cements is related to the principal problem of the MOC cement, the poor water resistance. Although in recent years, multiple research projects were conducted to improve the water resistance of magnesium cements, this is a major limitation that stopped the use of this cement at a larger scale. Regarding the composite building material, more research needs to be done to find an optimum mix proportion between wood waste and MOC cement, so that the strength of the new material can be comparable with one of the most common concrete made with ordinary Portland cements.

Overall, the use of wood waste and magnesium cement composites as building materials can provide a range of environmental benefits that make them a more sustainable choice than many traditional building materials.

## Figures and Tables

**Figure 1 materials-16-01944-f001:**
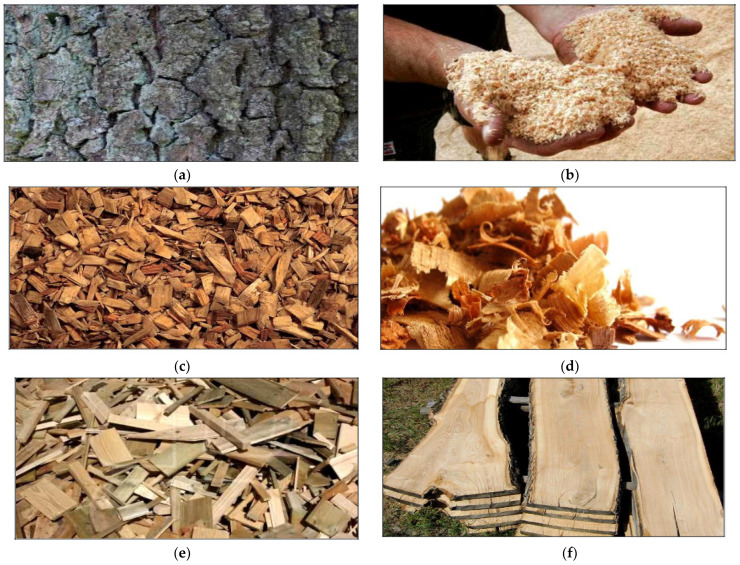
The main composition of wood waste [9]. (**a**) Wood bark; (**b**) wood sawdust; (**c**) wood chips: (**d**) wood chips: (**e**) wood cutouts; (**f**) rejected wood.

**Figure 2 materials-16-01944-f002:**
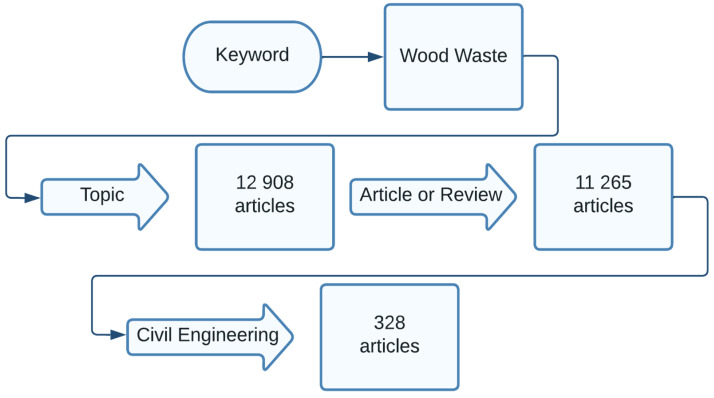
The data collection flow diagram.

**Figure 3 materials-16-01944-f003:**
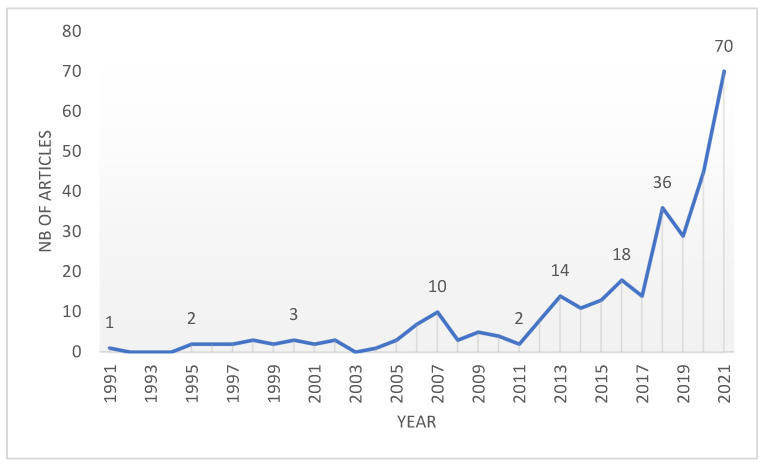
The evolution of the annual number of published papers having the “wood waste” topic.

**Figure 4 materials-16-01944-f004:**
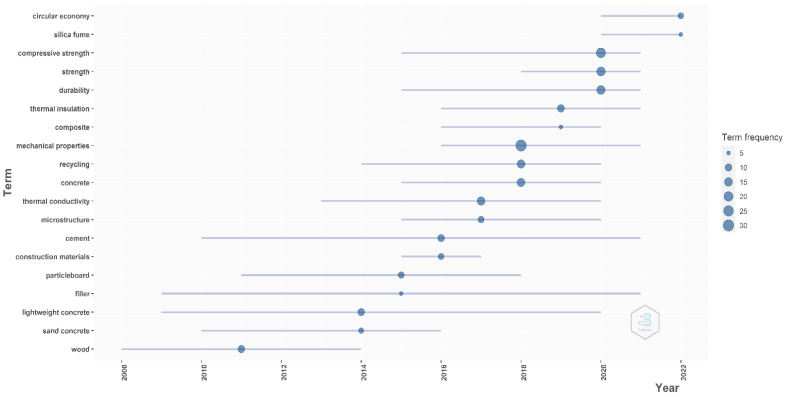
The trend topic of wood waste in the civil engineering field of research.

**Table 1 materials-16-01944-t001:** Classification of wood waste according to its origin [10].

Origin	Type	Class
Packaging	Pallets and boxes (untreated, no MDF)	1–2
	Pallets and boxes (with MDF/treated wood)	3
Construction/demolition	Wood from construction and rebuilding (untreated, no MDF)	1–2
	Old wood from demolition and rebuilding (with MDF/treated wood)	3
Furniture	Furniture (untreated, no fiberboard, and/or treated wood) 1–2	1–2
	Furniture (with fiberboard and/or treated wood)	3
	Furniture, upholstered	3
Others	Impregnated wood	4
	Composite Building materials from demolition	3
	Miscellaneous (items made out of plastic, glass, metal, cardboard)	3

## Data Availability

Not applicable.

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
