# Peer review of "A Review of the Environmental Benefits of Using Wood Waste and Magnesium Oxychloride Cement as a Composite Building Material"

_materials, 2023, doi:10.3390/ma16051944_

Round 1
Reviewer 1 Report (Previous Reviewer 1)
The revisions have sufficiently addressed the questions raised. But the reviewer think that this paper will be perfect, if the author will consider added some recent REF about the modifiers of MOC and wood-MOC. Currently, there are some REFs publised.
Author Response
Thank you very much for your review. I have carefully read your observations and implement them in the revised form of the paper.
I have attached a document where I provided a point-by-point response.

Reviewer 2 Report (Previous Reviewer 3)
Dear Author, although in the title of the paper you have specified that it is a review article, you should add a final paragraph to the Introduction Section, in which you reiterate that the purpose of the document is to offer an overview of the state of the art on the subject treated. You could also explain why this overview is necessary.
Also, in the spirit of a review paper, the title of Section 2 can be confusing to the reader. “Research Purpose” seems to refer to the research of which the article is a report, but this is not a research article. Please rephrase this title.
Author Response
Thank you very much for your review. I have carefully read your observations and implemented them in the revised form of the paper.
I have attached a document where I provided a point-by-point response.

Reviewer 3 Report (New Reviewer)
The theme is interesting, the work is well structured and well written in general.
However, there are aspects that can be improved:
· More references are needed throughout the text regarding studies with MOC, which is the theme of the work, especially being a review article.
The topic in 5 – in The mixing proportions of wood waste and MOC cement could be added more information about the range of percentages used and the main performance of it as resistance and durability.
· It would also be important to quantify studies on wood waste with MOC. For example, in Figure 4, which of these quantified studies refer to studies with MOC? Remember that the focus of the work is the use of MOC and not just wood waste.
Other comments:
1. Line 50 - Change the reference to a reference other than the author himself. This is a reference of an old study that is not directly related to the study and the introduction is not justified. It is abusive to introduce several references by the author himself.
5. In The mixing proportions of wood waste and MOC cement
Lines 393-394 – In this statement there are no references. The percentages are by volume or by mass? Clarify.
For an environmental impact assessment, it is important to know the amount of material per volume kg/m3. And compare this amount with the usual quantity in conventional concrete.
6. Line 411 – “ require less energy to produce”- How much? Compare with cement and lime.
4. Conclusions
conclusions need to be improved, they focus essentially on benefits and limitations are not mentioned.
References
Correctly format references according to the journal and align the text.
Check for errors along the text, example:
Line 523 – error in word susati-anble
Line 536 - error in word Efficeint
Author Response
Thank you very much for your review. I have carefully read your observations and implemented them in the revised form of the paper.
I have attached a document where I provided a point-by-point response.

Reviewer 4 Report (New Reviewer)
The results of “A review of the environmental benefits of using wood waste and magnesium oxychloride cement (MOC) as a composite 3 building material” are of potential interest. The introduction section provides sufficient background of past literatures. In the experimental result and discussion section, the results of literaturs are elaborately discussed. The conclusions are supported by the results. All the references are related to this research and also sufficient. However, the following corrections are to be carried out before the acceptance of the Manuscript.
1. Title: Remove (MOC) from Title.
2. Keywords: Remove MOC
3. Provide the results of literatures in table forms.
4. Provide the figures/images/charts of the literatures in result section while presenting
5. This manuscript reviewed 64 papers only. The author is advised to include more literatures.
6. The highlighting of sentences are found at many places of the manuscript. Correct it.
7. Correct the reference format.
Author Response
Thank you very much for your review. I have carefully read your observations and implemented them in the revised form of the paper.
I have attached a document where I provided a point-by-point response.

Round 2
Reviewer 3 Report (New Reviewer)
Thank you for the improvement of the aspects mentioned. The article is now more complete.
This manuscript is a resubmission of an earlier submission. The following is a list of the peer review reports and author responses from that submission.
Round 1
Reviewer 1 Report
This paper analyze the possibilities of further use of wood waste as a composite building material with magnesium oxychloride cement. Though some recent studies had been published in this filed, but many controversial issues about the wood-MOC could not be still solved, such as, the poor waterproof, lower strength. However, a good review requires many expertise included an intimate knowledge and a critical logic. Obviously, the reviewer thought the author's ideas cannot be seen, and the published results are not novel in the manuscript. The author needs to provide the more details of mix proportion of wood-MOCs, and compare and discuss its durability and physical property in this review. Therefore, the manuscript could not be accepted.
Reviewer 2 Report
Dear Author
The topic addressed in this paper is very interesting and actual. The concerns about the environmental impact of the construction industry are increasing and any paper that can is focusing on finding solutions to these issues is welcome.
The abstract of the paper has the main aspects needed in an article it brings the readers in the scope of the research and present the main findings. The introduction is good it allows readers to enter and familiarizes with the key concepts in this paper. The short literature review explains well the main concepts studied in this paper.
The materials and methods are well presented so it is easy to understand how the research was conducted. As a small recommendation you can add information about the search or MOC cements also and how are these related to the wood waste.
The presentation of composite materials made of wood waste and MOC cements is well written and has the necessary data to have an idea about the main research results in this field. The environmental benefits are well highlighted and in the discussion part all the information are well presented. The conclusion part is also well written and summarize all the results.
As a general conclusion, I consider that the paper deals with an interesting topic of research and can be accepted to the publications with small modification.
Reviewer 3 Report
Although this paper has been submitted as a scientific article, it does not contain any new scientific results. Rather, it is a review of the available bibliography. Therefore, it is suggested that Authors resubmit the paper as a review.